# RankUp: Boosting Semi-Supervised Regression with an Auxiliary Ranking Classifier

**Pin-Yen Huang**
Academia Sinica
Taipei, Taiwan
pyhuang97@gmail.com

**Szu-Wei Fu**
NVIDIA
Taipei, Taiwan
szuweif@nvidia.com

**Yu Tsao**
Academia Sinica
Taipei, Taiwan
yu.tsao@citi.sinica.edu.tw

## Abstract

State-of-the-art (SOTA) semi-supervised learning techniques, such as FixMatch and it's variants, have demonstrated impressive performance in classification tasks. However, these methods are not directly applicable to regression tasks. In this paper, we present RankUp, a simple yet effective approach that adapts existing semi-supervised classification techniques to enhance the performance of regression tasks. RankUp achieves this by converting the original regression task into a ranking problem and training it concurrently with the original regression objective. This auxiliary ranking classifier outputs a classification result, thus enabling integration with existing semi-supervised classification methods. Moreover, we introduce regression distribution alignment (RDA), a complementary technique that further enhances RankUp's performance by refining pseudo-labels through distribution alignment. Despite its simplicity, RankUp, with or without RDA, achieves SOTA results in across a range of regression benchmarks, including computer vision, audio, and natural language processing tasks. Our code and log data are open-sourced at `https://github.com/pm25/semi-supervised-regression`.

## 1 Introduction

The effectiveness of deep learning models heavily depends on the availability of labeled data. However, obtaining labeled data can be challenging in various scenarios. For instance, tasks like quality assessment often require multiple human annotators to label a single data point [15, 8, 19], resulting in a labor-intensive and time-consuming process. In domains where expert annotation is frequently required, such as medical data, the cost of acquiring labeled data can be extremely expensive [14, 21, 36]. To address these challenges, semi-supervised learning provides a powerful approach to reduce reliance on labeled data for training deep learning models [22, 34, 17, 24, 7]. By effectively leveraging the unlabeled data during model training, semi-supervised learning provides a means to enhance model performance while minimizing the need for extensive labeled data.

Recent state-of-the-art (SOTA) semi-supervised learning methods, such as FixMatch and its variants, use a confidence threshold technique to obtain high-quality pseudo-labels [22, 34, 27, 4, 31]. This approach involves generating pseudo-labels from unlabeled data and then filtering out those with low confidence scores. The model is then trained to produce predictions consistent with these high-quality pseudo-labels. Despite its success across various classification tasks, directly applying this technique to regression tasks encounters several challenges. First, unlike classification models, regression models typically lack confidence measures for their predictions, making the confidence threshold technique unfeasible. Additionally, one of the motivations behind using pseudo-labels is to increase the model's confidence in its predictions for unlabeled data, based on the low-density assumption [6, 25]. However, as there are no confidence measures in the predictions of regression models, relying on the low-density assumption and increasing the confidence of unlabeled data becomes unfeasible.

38th Conference on Neural Information Processing Systems (NeurIPS 2024).

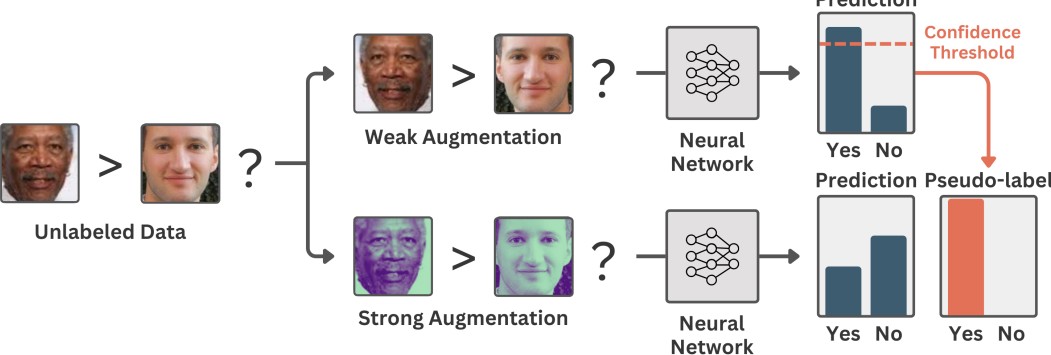

Figure 1: Illustration of using FixMatch on the Auxiliary Ranking Classifier (ARC). This diagram uses the age estimation task as an example, where the goal is to predict the age of a person in an image. The auxiliary ranking classifier transforms this task into a ranking problem by comparing two images to determine which person is older. (Image sourced from the UTKFace dataset [37]).

In this paper, we introduce *RankUp*, a simple yet effective semi-supervised regression framework that leverages existing semi-supervised classification methods. RankUp achieves this by using an auxiliary ranking classifier, which concurrently solves a ranking task alongside the original regression task. The ranking task is derived from the original regression problem, where the objective is to compare the labels of pairs of samples to determine their relative rank (i.e., which one is larger or smaller). Since ranking problem is a type of classification problem, existing semi-supervised classification methods can be applied to assist in training the auxiliary ranking classifier (see Fig. 1).

Our empirical results demonstrate that enhancing the performance of the auxiliary ranking classifier also improves the performance of the original regression task, as measured by metrics, such as mean absolute error (MAE) [29], coefficient of determination ($R^2$) [23], and Spearman rank correlation coefficient (SRCC) [33]. Furthermore, we show that applying existing semi-supervised classification methods to the auxiliary ranking classifier can effectively utilize unlabeled data, leading to further improvements in the classifier's performance. This improvement, in turn, translates to enhanced performance in the original regression task, showcasing the potential of applying semi-supervised classification techniques to enhance regression models.

One of the key advantages of using the auxiliary ranking classifier is its ability to enhance the ranking relationship of pseudo-labels. Building upon this effect, we propose a novel *Regression Distribution Alignment* (RDA) method, designed to further improve RankUp's performance by refining the distribution of regression pseudo-labels. RDA adjusts the distribution of these pseudo-labels to better align with the true underlying distribution of the unlabeled data. This approach assumes that the distributions of labeled and unlabeled data are similar, allowing us to estimate the distribution of the unlabeled data based on that of the labeled data distribution. This assumption holds true in many cases, especially when labeled data are randomly sampled from the same pool as the unlabeled data. By aligning these distributions, RDA improves the quality of the pseudo-labels, ultimately leading to better model performance when training with these refined pseudo-labels.

Our experimental results demonstrate that RankUp, even without RDA, achieves state-of-the-art (SOTA) results across a variety of regression datasets, including tasks in computer vision, audio, and natural language processing. Moreover, integrating RDA with RankUp provides an additional performance boost, leading to the highest performance observed in our experiments. For example, RankUp alone achieves at least a 13% improvement in MAE and a 28% improvement in $R^2$ compared to SOTA methods on the image age estimation dataset (UTKFace) with 50 labeled samples. The addition of RDA further boosts these results by an additional 6% and 7% in MAE and $R^2$, respectively. The empirical results of our experiments demonstrate that existing semi-supervised classification methods can be effectively leveraged to improve the performance of semi-supervised regression tasks. **These findings bridge the gap between future research in semi-supervised regression and classification, paving the way for further advancements in the field.**

## 2   Related Works

In this section, we review related research in semi-supervised learning. We categorize the literature into two groups: methods applicable to regression tasks, which will be discussed in Section 2.1, and methods applicable only to classification tasks, detailed in Section 2.2.

### 2.1   Semi-Supervised Regression

In semi-supervised regression, methods commonly rely on the smoothness assumption [6, 25], which suggests that nearby data points in the feature space should share similar labels. Consistency regularization is a popular technique employed to achieve this assumption. It encourages models to generate consistent predictions for slightly perturbed data.

For example, the $\Pi$-model [17] applies data augmentation to unlabeled data and minimizes the squared difference between the predictions of the augmented data and their original counterparts. Techniques like Mean Teacher [24] involve model-weight ensembling to align the predictions of the model with its ensemble counterpart. Similarly, UCVME [10] employs a bayesian neural network to ensure consistency in uncertainty predictions across co-trained models. Additionally, CLSS [11] utilizes contrastive learning to encourage features of similar labels to be closer together.

### 2.2   Semi-Supervised Classification

In semi-supervised classification, in addition to the smoothness assumption, another commonly relied-upon assumption is the low-density assumption [6, 25]. This assumption suggests that a classifier's decision boundary should ideally pass through low-density regions in the feature space. Pseudo-labeling [18] is a common approach used to achieve this assumption, where the highest probability class predictions on unlabeled data are utilized as pseudo-labels for training. By incorporating pseudo-labels, the model's confidence in predicting unlabeled data is increased, effectively pushing the decision boundary away from high-density regions towards low-density regions.

Recent SOTA semi-supervised learning methods combine pseudo-labeling with consistency regularization to achieve both the low-density and smoothness assumptions, leading to improved performance. For example, MixMatch [3] utilizes a mixup [35] technique and averages predictions from multiple augmented instances to ensure consistency, while also using a sharpening technique to boost prediction confidence on unlabeled data. Similarly, FixMatch [22] builds on this concept by generating high-quality pseudo-labels from weakly augmented unlabeled data using a confidence threshold and enforcing consistency between weakly and strongly augmented versions of the same input.

Despite the success of these methods on classification tasks. The low-density assumption doesn't directly translate to regression tasks, as regression models lack explicit confidence measures and decision boundaries like those in classification models. As a result, existing semi-supervised learning methods based on the low-density assumption cannot be directly applied in regression settings.

## 3   Method

The proposed framework, RankUp, introduces two additional components: ARC and RDA. The design of ARC is inspired by RankNet [5]. To provide a clear understanding of the ARC's implementation, we first present background information on RankNet in Section 3.1. Subsequently, we will detail the implementation of ARC in Section 3.2 and introduce RDA in Section 3.3. Furthermore, we propose a warm-up scheme and techniques for reducing the computational time of RDA in Sections 3.4 and 3.5, respectively. Lastly, we outline the complete RankUp framework in Section 3.6.

### 3.1   Background: RankNet

RankNet [5] is a deep learning model designed to predict the relevance scores of documents. The core idea behind RankNet is the use of a pairwise ranking loss. It compares two samples and predicts their relative ranking (i.e., which document is more relevant). This approach effectively transforms the relevance score prediction task into a pairwise classification problem. In the following, we will provide a detailed explanation of how the pairwise ranking prediction is performed and how the corresponding loss is calculated.

**Pairwise Ranking Prediction.** The output of RankNet is a single scalar value indicating the ranking score of the sample, where a higher score indicates greater relevance. To obtain the pairwise ranking prediction, two samples are fed separately into the model to get their respective ranking scores. The difference between these scores is then passed through a sigmoid function, which generates a prediction in the range [0, 1]. This prediction indicates the likelihood that the first sample is more relevant than the second. If the output is greater than 0.5, the model predicts that the first sample has higher relevance; if the output is less than 0.5, the second sample is considered more relevant. Mathematically, for two samples, $x_i$ and $x_j$, and the RankNet model $g$, the formula to obtain the pairwise ranking prediction $p_{ij}$ is as follows:

$$p_{ij} = \text{sigmoid}(g(x_i) - g(x_j)) = \frac{1}{1 + e^{-(g(x_i) - g(x_j))}} \tag{1}$$

Here, $g(x_i)$ and $g(x_j)$ represent the ranking scores for samples $x_i$ and $x_j$, respectively. A higher value of $p_{ij}$ indicates a higher likelihood that the ranking of $x_i$ will be higher than that of $x_j$.

**Pairwise Ranking Loss.** The pairwise ranking loss is calculated by comparing the model's predicted pairwise ranking $p_{ij}$ with the ground truth label $y_{ij}$. The label $y_{ij}$ indicates the true relative ranking between samples $x_i$ and $x_j$. Specifically, $y_{ij} = 1$ indicates that sample $x_i$ is ranked higher than sample $x_j$, $y_{ij} = 0$ indicates that sample $x_i$ is ranked lower than $x_j$, and $y_{ij} = 0.5$ suggests the two samples are equally ranked. Since this is fundamentally a binary classification task, the pairwise ranking loss is calculated using the cross-entropy loss function. Mathematically, the pairwise ranking loss for a batch of data is defined as follows:

$$\ell_{ranknet} = \frac{1}{N^2} \sum_{i=1}^{N} \sum_{j=1}^{N} \text{CE}(y_{ij}, \, p_{ij}) \tag{2}$$

Here, $N$ denotes the batch size, CE is the cross-entropy loss function, $p_{ij}$ is the predicted pairwise ranking between samples $x_i$ and $x_j$, and $y_{ij}$ is the corresponding ground truth label. The loss iterates through all possible pairs of samples in the batch to calculate the average loss for the entire batch.

## 3.2 Auxiliary Ranking Classifier (ARC)

The *Auxiliary Ranking Classifier* (ARC) is designed to solve a ranking task alongside the primary regression task. It can be easily integrated into existing regression model architectures like ResNet [13], BERT [12], or Whisper [20]. ARC is implemented as an additional output layer that shares the same hidden layers with the original regression model. This transforms the model into a multi-task architecture with two output tasks: the original output header continues to provide the regression output, while ARC generates a ranking score for the sample.

The core idea behind ARC is to transform the original regression task into a multi-class classification problem, allowing existing semi-supervised classification methods to assist in its training. To achieve this, ARC's design is inspired by RankNet, which can effectively convert the regression task into a binary classification task. However, since a multi-class classification output is required, we introduce two key modifications to RankNet to adapt it for this purpose:

1. The scalar output value of RankNet is changed to a two-class output, where each output class indicates which sample in a pair has a relatively greater ranking score.

2. The sigmoid function is replaced with softmax, which converts the model's output into a multi-class classification probability distribution.

Specifically, for the auxiliary ranking classifier $r$, which outputs a two-class output, the formula to obtain the pairwise ranking prediction $\hat{p}_{ij}$ of two data samples, $x_i$ and $x_j$, is as follows:

$$\hat{p}_{ij} = \text{softmax}(r(x_i) - r(x_j)) \tag{3}$$

Here, $\hat{p}_{ij}$ represents a two-class prediction that indicates which sample in the pair has a relatively higher regression label. The loss calculation for ARC remains the same as described in Equation 2.

This output format enables the integration of existing semi-supervised classification methods. We utilize FixMatch [22] as the semi-supervised classification technique for training ARC. An illustrative example of applying FixMatch to ARC can be found in Figure 1, with further details in Algorithm 1.

---

**Algorithm 1** Auxiliary Ranking Classifier (with FixMatch)

---

**Input:** Labeled batch $X = \{ (x_i, y_i) \}_{i=1}^{N_{lb}}$, unlabeled batch $U = \{ u_i \}_{i=1}^{N_{ulb}}$, confidence threshold $\tau$, unlabeled loss weight $\omega_{ulb}$, weak augmentation $A_w$, strong augmentation $A_s$

1: $\ell_{lb} = \frac{1}{(N_{lb})^2} \sum_{i=1}^{N_{lb}} \sum_{j=1}^{N_{lb}} \text{CE}\Big( \text{softmax}\big(r(A_w(x_i)) - r(A_w(x_j))\big), \ \mathbb{1}\{y_i > y_j\} \Big)$ { Compute cross-entropy labeled loss }
2: $\ell_{ulb} = 0$ { Initialize unlabeled loss }
3: **for** $i = 1$ **to** $N_{ulb}$ **do**
4:     **for** $j = 1$ **to** $N_{ulb}$ **do**
5:         $\hat{p}_{ij}^w = \text{softmax}(r(A_w(u_i)) - r(A_w(u_j)))$ { Predict weak pairwise ranking }
6:         $\hat{p}_{ij}^s = \text{softmax}(r(A_s(u_i)) - r(A_s(u_j)))$ { Predict strong pairwise ranking }
7:         $\ell_{ulb} = \ell_{ulb} + \mathbb{1}\{\max(\hat{p}_{ij}^w) > \tau\} \text{CE}(\arg\max(\hat{p}_{ij}^w), \hat{p}_{ij}^s)$ { Accumulate unlabeled loss }
8:     **end for**
9: **end for**
10: $\ell_{ulb} = \frac{1}{(N_{ulb})^2} \ell_{ulb}$ { Average unlabeled loss }
11: **return** $\ell_{\text{arc}} = \ell_{lb} + \omega_{ulb} \cdot \ell_{ulb}$

---

### 3.3 Regression Distribution Alignment (RDA)

Distribution alignment is a commonly used technique in semi-supervised classification [2, 16, 28, 4], where pseudo-labels are refined by aligning their distribution with that of the labeled data. Training semi-supervised models with these refined pseudo-labels can lead to performance improvements. However, existing distribution alignment methods are designed for classification tasks involving discrete label distributions, making them unsuitable for regression settings. Moreover, applying distribution alignment to ARC is impractical, as the two output classes always have equal proportions. To address these challenges, we propose *Regression Distribution Alignment* (RDA), enabling the direct application of distribution alignment to regression tasks.

The RDA process involves three key steps: (1) extracting the labeled data distribution, (2) generating the pseudo-label distribution, and (3) aligning the pseudo-label distribution with the labeled data distribution. These steps correspond to the orange, blue, and yellow parts of Figure 2, respectively.

**Step 1: Extracting the Labeled Data Distribution.** The labeled data is sorted according to its label values. To ensure a one-to-one correspondence with the pseudo-labels, the labeled data distribution must contain the same number of data points as the pseudo-label set. We use linear interpolation to extend the labeled distribution to match the size of the pseudo-labels.

**Step 2: Generating the Pseudo-Label Distribution.** The model generates pseudo-labels for all the unlabeled data. These pseudo-labels are then sorted by their values, either in ascending or descending order, as long as the sorting direction is consistent with that of the labeled data distribution.

**Step 3: Aligning the Distributions.** Once both distributions have been sorted and resized to the same size, the alignment is performed by replacing each pseudo-label value with its corresponding value from the labeled data distribution.

In each training iteration, RDA is applied to refine the pseudo-labels. The loss is then computed between the model's predictions and the RDA-aligned pseudo-labels to minimize their difference. For an unlabeled data point $u_i$ with its corresponding regression prediction $\hat{y}_i$ and RDA-aligned pseudo-label $\tilde{y}_i$, the RDA loss $\ell_{\text{rda}}$ for a batch of unlabeled data is defined as:

$$\ell_{\text{rda}} = \frac{1}{N_{ulb}} \sum_{i=1}^{N_{ulb}} L_{reg}\left( \hat{y}_i, \ \tilde{y}_i \right) \tag{4}$$

Here, $N_{ulb}$ denotes the batch size of unlabeled data, $L_{reg}$ represents a regression loss function (e.g., MAE, MSE).

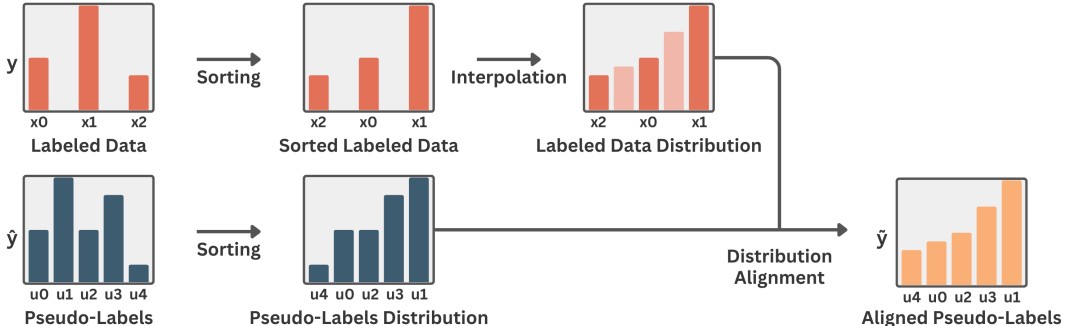

Figure 2: Illustration of RDA: This example includes three labeled data pairs $\{(x_i,\ y_i)\}_{i=0}^2$ and five unlabeled data points with corresponding pseudo-labels $\{(u_i,\ \hat{y}_i)\}_{i=0}^4$. Each data pair is represented by a single bar in the graph. The x-axis indicates the sample indices, while the y-axis represents their corresponding regression label values. The orange bars demonstrate the process of obtaining the labeled data distribution, the blue bars illustrate how the pseudo-label distribution is formed, and the yellow bars show the aligned pseudo-labels after applying RDA.

The design of RDA is based on two key assumptions. First, it assumes that the distributions of labeled and unlabeled data are similar, which is often true since labeled data is typically randomly sampled from the unlabeled pool. Second, it assumes that the ranking of the pseudo-labels is reasonably accurate. Integrating RDA with ARC can reinforce this assumption, as ARC enhances the ranking relationships of pseudo-labels. Both assumptions are crucial for ensuring that RDA works properly.

### 3.4 Warm-Up Scheme for RDA

In the early stages of training, the pseudo-label rankings may be poorly predicted, which can degrade the quality of the pseudo-labels refined through RDA. To address this, we introduce a linear warm-up scheme to stabilize the RDA process. The adjusted RDA loss, $\ell'_{\text{rda}}$, is defined as follows:

$$\ell'_{\text{rda}} = \min\left(\frac{iter}{\alpha_{\text{warm}}},\ 1.0\right) \cdot \ell_{\text{rda}} \tag{5}$$

Here, $iter$ denotes the current training iteration, and $\alpha_{\text{warm}}$ is a hyperparameter that controls the duration of the warm-up phase. The $\min$ function ensures that the warm-up factor does not exceed 1.0, smoothly transitioning the model toward the full effect of $\ell_{\text{rda}}$.

### 3.5 Reducing Computational Time of RDA

Applying RDA can be computationally expensive, as it requires inference all unlabeled data and sorting all pseudo-labels at every training iteration. This significantly increases the computational load compared to the original training process, especially when dealing with a large volume of unlabeled data, making the implementation of RDA impractical. To mitigate this challenge, we propose several techniques aimed at reducing the computational burden of RDA.

**Pseudo-label table.** RDA creates a table of the same size as the unlabeled dataset. This table stores the model's predicted pseudo-labels for each instance of unlabeled data. For each training iteration, the model generates new pseudo-labels, which are stored and updated within this table. This approach eliminates the need to rerun inference on all unlabeled data when applying RDA, as it only requires a simple lookup from the pseudo-label table.

**Applying RDA only every $T$ steps.** To further reduce computational costs, RDA is applied only every $T$ steps, where $T$ is a hyperparameter. This is achieved by creating a second table of the same size as the unlabeled dataset, which stores the previously aligned results of the pseudo-labels generated by applying RDA. Between RDA updates, the model uses these stored aligned pseudo-labels, thereby avoiding the need to run RDA in every iteration. This strategy effectively reduces the computational cost associated with the RDA process to $1/T$.

### 3.6 Putting It All Together - RankUp

We introduce the term *RankUp* to describe our proposed framework, which integrates two key components: ARC and RDA. The use of RDA is optional, depending on whether its underlying assumptions are satisfied. The final loss for RankUp is a combination of the regression loss and the ARC loss. The regression loss consists of the original labeled regression loss plus the unlabeled RDA loss. Specifically, the RankUp loss $\ell_{\text{rankup}}$ is defined as follows:

$$\ell_{\text{rankup}} = (\ell_{\text{reg}} + \omega_{\text{rda}} \cdot \ell'_{\text{rda}}) + (\omega_{\text{arc}} \cdot \ell_{\text{arc}}) \tag{6}$$

In this equation, $\ell_{\text{reg}}$ represents the loss from the original labeled regression task, while $\ell'_{\text{rda}}$ denotes the RDA loss, as defined in Equation 5. The hyperparameter $\omega_{\text{rda}}$ controls the weight of the RDA loss. If RDA is not employed, the term $\omega_{\text{rda}} \cdot \ell'_{\text{rda}}$ can be excluded from the equation. The term $\ell_{\text{arc}}$ corresponds to the loss from the ARC module, as detailed in Algorithm 1. Additionally, the hyperparameter $\omega_{\text{arc}}$ regulates the weight of the ARC loss.

## 4 Experiments

In this section, we evaluate RankUp's performance across various tasks. The experimental settings are described in Section 4.1. The main results for RankUp under different label configurations are presented in Section 4.2, while Section 4.3 provides additional results on audio and text datasets. Section 4.4 explores the use of alternative semi-supervised classification methods in place of FixMatch. Lastly, we discuss potential reasons why the smoothness and low-density assumptions are also effective for regression tasks in Section 4.5.

### 4.1 Settings

**Evaluation Metrics.** We use three evaluation metrics: MAE, $R^2$, and SRCC, to assess the performance of semi-supervised regression methods. MAE measures the average absolute difference between the model's predictions and the actual values. The $R^2$ score indicates the proportion of variance in the data explained by the model. SRCC evaluates the correlation between the predicted rankings and the actual rankings.

**Evaluation Robustness.** To ensure the reliability of our evaluation results, each experiment is executed three times using fixed random seeds (0, 1, and 2). We report both the mean and standard deviation of each metric.

**Fair Comparison.** To ensure a fair comparison between our proposed methods and related works, we implement and evaluate all methods within the same codebase. Specifically, we adapt the popular semi-supervised classification framework USB [26], modifying it for regression tasks to implement both our proposed methods and related works. Weak augmentation is applied consistently to the labeled data across all semi-supervised and supervised methods. For specific details on the modifications made to USB, please refer to Appendix A.5. The code and full training logs of the experiments presented in this paper are open-sourced at `https://github.com/pm25/semi-supervised-regression`.

**Hyperparameters.** We use the hyperparameters of USB as the base for fine-tuning. We first fine-tune the hyperparameters in the supervised baseline setting and find the hyperparameters that lead to lowest MAE score. These same hyperparameters are then applied to all semi-supervised regression methods to ensure a fair comparison. Only the additional hyperparameters specific to each semi-supervised method are further tuned. For more details on the hyperparameters, please refer to Appendix A.13.

**Base Model.** The base model used in our experiments varies depending on the data type. For image data, we use Wide ResNet-28-2 [32], which is not pre-trained. For audio data, we use the pre-trained Whisper-base [20], and for text data, we use the pre-trained Bert-Small [12].

**Dataset.** To simulate the semi-supervised setting, we randomly sample a portion of the dataset as labeled data, treating the remainder as unlabeled. To evaluate performance, we use three diverse datasets: UTKFace [37], an image age estimation dataset; BVCC [8], an audio quality assessment dataset; and Yelp Review [1], a text sentiment analysis (opinion mining) dataset. For more detailed information about these datasets, please refer to Appendix A.11.

Table 1: Comparison of RankUp with and without RDA against other methods on the UTKFace dataset, evaluated under two settings: 50 and 250 labeled samples, with the remaining images treated as unlabeled. The original UTKFace dataset comprises 18,964 training images.

| | UTKFace (Image Age Estimation) | | | | | |
| | Labels = 50 | | | Labels = 250 | | |
| | MAE↓ | $R^2$↑ | SRCC↑ | MAE↓ | $R^2$↑ | SRCC↑ |
|---|---|---|---|---|---|---|
| Supervised | $14.13_{\pm0.56}$ | $0.090_{\pm0.092}$ | $0.371_{\pm0.071}$ | $9.42_{\pm0.16}$ | $0.540_{\pm0.014}$ | $0.712_{\pm0.010}$ |
| Π-Model | $13.82_{\pm1.02}$ | $0.100_{\pm0.086}$ | $0.387_{\pm0.092}$ | $9.45_{\pm0.30}$ | $0.534_{\pm0.030}$ | $0.706_{\pm0.015}$ |
| Mean Teacher | $13.92_{\pm0.20}$ | $0.127_{\pm0.037}$ | $0.423_{\pm0.023}$ | $8.85_{\pm0.25}$ | $0.586_{\pm0.020}$ | $0.745_{\pm0.013}$ |
| CLSS | $13.61_{\pm0.92}$ | $0.138_{\pm0.101}$ | $0.447_{\pm0.074}$ | $9.10_{\pm0.15}$ | $0.586_{\pm0.016}$ | $0.737_{\pm0.014}$ |
| UCVME | $13.49_{\pm0.95}$ | $0.157_{\pm0.110}$ | $0.412_{\pm0.127}$ | $8.63_{\pm0.17}$ | $0.626_{\pm0.006}$ | $0.767_{\pm0.007}$ |
| MixMatch | $11.44_{\pm0.45}$ | $0.401_{\pm0.028}$ | $0.674_{\pm0.035}$ | $7.95_{\pm0.15}$ | $0.692_{\pm0.013}$ | $0.832_{\pm0.008}$ |
| RankUp (Ours) | $9.96_{\pm0.62}$ | $0.514_{\pm0.043}$ | $0.703_{\pm0.019}$ | $7.06_{\pm0.11}$ | $0.751_{\pm0.011}$ | $0.835_{\pm0.008}$ |
| RankUp + RDA (Ours) | $\mathbf{9.33_{\pm0.54}}$ | $\mathbf{0.552_{\pm0.041}}$ | $\mathbf{0.770_{\pm0.009}}$ | $\mathbf{6.57_{\pm0.18}}$ | $\mathbf{0.782_{\pm0.012}}$ | $\mathbf{0.856_{\pm0.005}}$ |
| Fully-Supervised | $4.85_{\pm0.01}$ | $0.875_{\pm0.000}$ | $0.910_{\pm0.001}$ | $4.85_{\pm0.01}$ | $0.875_{\pm0.000}$ | $0.910_{\pm0.001}$ |

## 4.2 Main Results

To evaluate the performance of RankUp under different labeled data settings, we conducted experiments using the UTKFace dataset with 50 and 250 labeled samples. We tested two configurations of RankUp: one incorporating the RDA (RankUp + RDA) and the other without it (RankUp). Their performance was compared against other semi-supervised regression methods, with MixMatch specifically representing the consistency regularization component of the approach. Additionally, we included a supervised setting that used only the labeled data **without** incorporating any unlabeled data during training, as well as a fully-supervised setting that used **all** available data (both labeled and unlabeled), assuming the unlabeled data had known true labels. We also conducted experiments with a 2000 labeled samples setting; however, due to space limitations, the results for this configuration can be found in Appendix A.2.

The results are presented in Table 1. We observed that RankUp (without RDA) consistently outperforms existing semi-supervised regression methods, especially when the amount of labeled data is scarce. Specifically, in the 50-label setting, RankUp achieves at least a 12.9% improvement in MAE, a 28.2% improvement in $R^2$, and a 4.3% improvement in SRCC compared to other semi-supervised regression methods. In the 250-label setting, RankUp shows at least an 11.2% improvement in MAE, an 8.5% improvement in $R^2$, and a 0.4% improvement in SRCC.

Furthermore, the integration of RDA with RankUp further enhances the performance of RankUp. Specifically, in the 50-label setting, RankUp + RDA achieves an additional 6.3% improvement in MAE, a 7.4% improvement in $R^2$, and a 9.5% improvement in SRCC compared to RankUp alone. Similarly, in the 250-label setting, RankUp + RDA achieves an additional 6.9% improvement in MAE, a 4.1% improvement in $R^2$, and a 2.5% improvement in SRCC relative to RankUp.

These empirical results demonstrate the effectiveness of RankUp and RDA across different labeled settings. Another notable observation is that RankUp + RDA in the 50-label setting outperforms the supervised model that utilizes five times the labeled data (in the 250-label setting) across all three metrics. Specifically, RankUp + RDA achieves a 1.0% improvement in MAE, a 2.2% improvement in $R^2$, and an 8.1% improvement in SRCC while using only one-fifth of the labeled data, demonstrating its effectiveness in reducing labeling costs.

## 4.3 Additional Results on Audio and Text Data

To further assess the performance of RankUp across different data types and tasks, we evaluated it on the BVCC and Yelp Review datasets using 250 labeled samples. The results are presented in Table 2. The table demonstrates that RankUp also consistently outperforms existing semi-supervised regression methods on both audio and text datasets. Specifically, on the BVCC dataset, RankUp (without RDA) achieves at least a 5.6% improvement in MAE, a 6.3% improvement in $R^2$, and a 0.3% improvement in SRCC compared to other semi-supervised regression methods. On the Yelp

Table 2: Comparison of RankUp with and without RDA against other methods on the BVCC and Yelp Review datasets, evaluated under the 250-labeled samples setting. The BVCC dataset consists of 4,975 training audio samples, while the Yelp Review dataset contains 250,000 training text comments.

| | BVCC (Audio Quality Assessment) | | | Yelp Review (NLP Opinion Mining) | | |
| --- | --- | --- | --- | --- | --- | --- |
| | Labels = 250 | | | Labels = 250 | | |
| | MAE$\downarrow$ | $R^2\uparrow$ | SRCC$\uparrow$ | MAE$\downarrow$ | $R^2\uparrow$ | SRCC$\uparrow$ |
| Supervised | $0.533_{\pm0.006}$ | $0.490_{\pm0.018}$ | $0.741_{\pm0.009}$ | $0.723_{\pm0.023}$ | $0.566_{\pm0.019}$ | $0.769_{\pm0.010}$ |
| Π-Model | $0.534_{\pm0.008}$ | $0.489_{\pm0.021}$ | $0.740_{\pm0.009}$ | $0.730_{\pm0.024}$ | $0.565_{\pm0.019}$ | $0.769_{\pm0.009}$ |
| Mean Teacher | $0.532_{\pm0.006}$ | $0.492_{\pm0.018}$ | $0.742_{\pm0.008}$ | $0.730_{\pm0.024}$ | $0.565_{\pm0.019}$ | $0.769_{\pm0.009}$ |
| CLSS | $0.499_{\pm0.010}$ | $0.534_{\pm0.027}$ | $0.748_{\pm0.008}$ | $0.721_{\pm0.010}$ | $0.543_{\pm0.011}$ | $0.748_{\pm0.002}$ |
| UCVME | $0.498_{\pm0.003}$ | $0.553_{\pm0.011}$ | $0.774_{\pm0.008}$ | $0.775_{\pm0.006}$ | $0.540_{\pm0.005}$ | $0.763_{\pm0.005}$ |
| MixMatch | $0.597_{\pm0.017}$ | $0.353_{\pm0.044}$ | $0.626_{\pm0.031}$ | $0.886_{\pm0.004}$ | $0.381_{\pm0.008}$ | $0.660_{\pm0.004}$ |
| RankUp (Ours) | $0.470_{\pm0.012}$ | $0.588_{\pm0.028}$ | $0.776_{\pm0.010}$ | $0.661_{\pm0.018}$ | $0.645_{\pm0.013}$ | $\mathbf{0.829}_{\pm0.002}$ |
| RankUp + RDA (Ours) | $\mathbf{0.463}_{\pm0.013}$ | $\mathbf{0.598}_{\pm0.027}$ | $\mathbf{0.783}_{\pm0.011}$ | $\mathbf{0.632}_{\pm0.009}$ | $\mathbf{0.651}_{\pm0.007}$ | $0.810_{\pm0.005}$ |
| Fully-Supervised | $0.351_{\pm0.003}$ | $0.764_{\pm0.002}$ | $0.874_{\pm0.001}$ | $0.418_{\pm0.003}$ | $0.799_{\pm0.002}$ | $0.896_{\pm0.001}$ |

Table 3: Comparison of using different semi-supervised classification methods for training RankUp's ARC component. Results are evaluated on the UTKFace dataset with a setting of 250 labeled samples.

| | MAE$\downarrow$ | $R^2\uparrow$ | SRCC$\uparrow$ |
| --- | --- | --- | --- |
| None | $9.42_{\pm0.16}$ | $0.540_{\pm0.014}$ | $0.712_{\pm0.010}$ |
| Supervised | $9.03_{\pm0.09}$ | $0.588_{\pm0.018}$ | $0.746_{\pm0.008}$ |
| Π-Model | $8.81_{\pm0.11}$ | $0.591_{\pm0.013}$ | $0.751_{\pm0.012}$ |
| Mean Teacher | $8.76_{\pm0.13}$ | $0.607_{\pm0.018}$ | $0.750_{\pm0.005}$ |
| FixMatch | $\mathbf{7.06}_{\pm0.11}$ | $\mathbf{0.751}_{\pm0.011}$ | $\mathbf{0.835}_{\pm0.008}$ |

Review dataset, RankUp shows at least a 8.3% improvement in MAE, a 14.2% improvement in $R^2$, and a 7.8% improvement in SRCC relative to other semi-supervised regression methods.

Moreover, integrating RDA with RankUp further enhances RankUp's performance on both datasets. In the BVCC dataset, RankUp + RDA achieves an additional 1.5% improvement in MAE, a 1.7% improvement in $R^2$, and a 0.9% improvement in SRCC. In the Yelp Review dataset, RankUp + RDA shows an additional 4.4% improvement in MAE and a 0.9% improvement in $R^2$. However, in the Yelp Review dataset, RankUp + RDA did not improve SRCC compared to RankUp without RDA, resulting in a 2.3% decrease in SRCC. This decline is likely due to the limited distinct label values (only five distinct values) in the Yelp Review dataset; applying RDA may cause many pseudo-labels to align with the same value, thereby disrupting the ranking relationships and leading to a degradation in SRCC performance.

## 4.4 Analysis of Different Semi-Supervised Classification Methods on ARC

To better understand the effect of RankUp's ARC component and its ability to utilize the unlabeled data, we analyzed ARC by training it with various semi-supervised classification methods. In this evaluation, we did not apply RDA to isolate its effect on ARC. Additionally, we compared these methods against a baseline with no ARC (denoted as "None") and a supervised setting where only labeled data was used to train ARC (denoted as "Supervised").

The results are presented in Table 3. The table indicates that using an ARC without any unlabeled data can already improve performance, with a 4.1% improvement in MAE, an 8.9% improvement in $R^2$, and a 4.8% improvement in SRCC compared to the "None" setting. This suggests the beneficial impact of training ARC concurrently with the original regression head, even without leveraging unlabeled data.

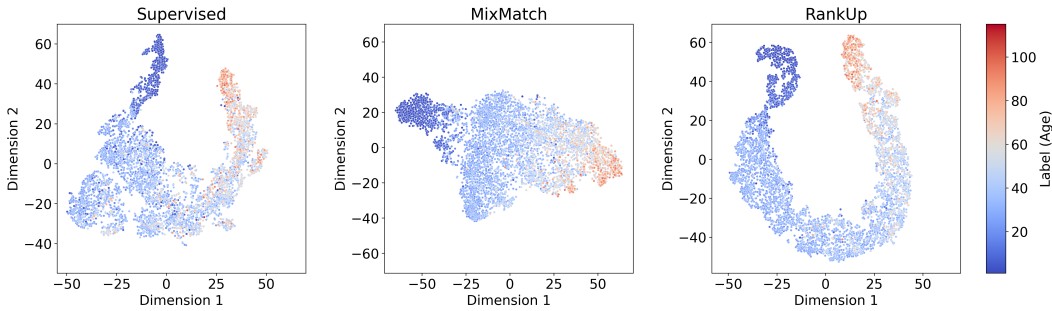

Figure 3: Comparison of t-SNE visualizations of feature representations for different semi-supervised regression methods on evaluation data. The supervised model is displayed on the left, MixMatch is in the center, and RankUp (without RDA) is shown on the right.

Furthermore, using different semi-supervised classification methods to utilize the unlabeled data further boost performance compared to using only labeled data. Specifically, we tested the Π-Model, Mean Teacher, and FixMatch, all of which demonstrated improvements over the "Supervised" setting across MAE, $R^2$, and SRCC metrics. Among these methods, FixMatch achieved the best results, showing a 21.8% improvement in MAE, a 27.7% improvement in $R^2$, and an 11.9% improvement in SRCC compared to the "Supervised" setting. This highlights the effectiveness of leveraging unlabeled data and semi-supervised classification techniques to improve ARC and RankUp's overall performance.

### 4.5 Understanding Smoothness and Low-Density Assumptions in Regression

The low-density assumption is crucial for understanding the effectiveness of semi-supervised learning methods. However, it does not directly apply to regression tasks due to the absence of decision boundaries and confidence measures. In this section, we explore why RankUp performs well in regression tasks by leveraging semi-supervised classification techniques that utilize the low-density assumption. This understanding can broaden our perspective on these assumption.

We adopt a broader interpretation of the smoothness and low-density assumptions. Rather than viewing them solely in the context of classification, we interpret the smoothness assumption as an effort to **group features with similar labels together**, while the low-density assumption aims to **separate features with dissimilar labels**. These perspectives align with RankUp's approach. By training ARC with pseudo-labels, the model is encouraged to have greater confidence in the pairwise ranking predictions of the unlabeled data, thus pushing the features with dissimilar pseudo-labels further apart. Additionally, ensuring consistent predictions between weakly and strongly augmented data assists in grouping features with similar labels. The t-SNE visualization demonstrated in Figure 3 supports this claim, showing that within RankUp, similar labels are closer together, while dissimilar labels are pushed further apart in the feature space.

## 5 Conclusion

Recent advancements in semi-supervised learning have achieved impressive results across various classification tasks; however, these methods are not directly applicable to regression tasks. In this work, we investigate the potential of leveraging existing semi-supervised classification techniques for regression tasks. We propose a novel framework, RankUp, which introduces two key components: the Auxiliary Ranking Classifier (ARC) and Regression Distribution Alignment (RDA). The empirical results of our experiments demonstrate the effectiveness of our methods across various labeled data settings (50, 250, and 2000 labeled samples) and different types of datasets (image, audio, and text). These findings show that semi-supervised classification techniques can be effectively adapted to regression tasks, bridging the gap between research in semi-supervised regression and classification, and paving the way for more advanced research in this area.

# 6  Acknowledgments

I sincerely appreciate everyone who made this work possible. I am especially thankful to my coauthors, Szu-Wei Fu and Prof. Yu Tsao for their mentorship and our weekly discussions; their insights and comments have been invaluable in shaping this research. I am also very grateful for my experience at CLLab and the guidance of Hsuan-Tien Lin, which sparked my interest in weakly-supervised learning and greatly influenced my research mindset and approach. My heartfelt thanks go to my family for their unwavering support throughout my academic journey. I also thank my friend Chi-Chang, whose discussions helped me develop a deeper understanding of how to approach research. Lastly, I want to thank Chia-Ling for her continuous support and encouragement during the development of this research. This work would not have been possible without all these individuals. Thank you!

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

# A Appendix / supplemental material

## A.1 Limitations

One of the limitations of RDA is that it relies on two key assumptions: the labeled and unlabeled data have similar label distributions, and the ranking of the pseudo-labels is accurate. If either of these assumptions is not met, the performance of RDA can be significantly impacted.

Another limitation of RDA is that it may not perform well in tasks where distinct label values are few. As shown in Table 2, in the Yelp Review dataset, SRCC decreases when RankUp is combined with RDA compared to using RankUp alone. This is likely because the limited number of distinct label values causes many pseudo-labels to align to the same value, disrupting the ranking relationships.

Despite these limitations, RDA is still a powerful component that can work with RankUp to further boost its performance. However, users should carefully evaluate their dataset characteristics to determine whether applying RDA is an appropriate choice.

## A.2 Analysis on UTKFace 2000 Labeled Setting

In Table 1, we present the results of RankUp against other methods using 50 and 250 labeled samples settings on the UTKFace dataset. Here, we further investigate the performance of RankUp in a 2000-label setting, with the results shown in Table 4. Notably, RankUp continues to exhibit performance improvements over other semi-supervised regression methods in a 2000-label setting. Specifically, RankUp (without RDA) demonstrates a 7.0% improvement in MAE, a 1.7% improvement in $R^2$, and a 0.5% improvement in SRCC. Meanwhile, RankUp with RDA demonstrates an additional 1.8% improvement in MAE, a 0.7% improvement in $R^2$, and a 0.3% improvement in SRCC compared to RankUp alone. Comparing the results in Table 1 (50 and 250 labeled samples) with those in Table 4 (2000 labeled samples), we observe that RankUp with or without RDA continue to outperform other methods as the number of labeled samples increases. However, as expected, the advantages of semi-supervised learning diminish with the availability of more labeled training data.

Table 4: Comparison of RankUp with and without RDA against other methods on the UTKFace dataset with 2000 labeled samples.

| | UTKFace (Image Age Estimation) | | |
| --- | --- | --- | --- |
| | Labels = 2000 | | |
| | MAE$\downarrow$ | $R^2\uparrow$ | SRCC$\uparrow$ |
| Supervised | $6.28_{\pm0.06}$ | $0.794_{\pm0.004}$ | $0.862_{\pm0.001}$ |
| Π-Model | $6.31_{\pm0.10}$ | $0.790_{\pm0.006}$ | $0.860_{\pm0.003}$ |
| Mean Teacher | $6.29_{\pm0.03}$ | $0.793_{\pm0.004}$ | $0.862_{\pm0.001}$ |
| CLSS | $6.29_{\pm0.01}$ | $0.794_{\pm0.003}$ | $0.862_{\pm0.001}$ |
| UCVME | $5.90_{\pm0.07}$ | $0.821_{\pm0.007}$ | $0.877_{\pm0.002}$ |
| MixMatch | $6.03_{\pm0.07}$ | $0.824_{\pm0.004}$ | $0.883_{\pm0.002}$ |
| RankUp (Ours) | $5.61_{\pm0.07}$ | $0.838_{\pm0.003}$ | $0.887_{\pm0.003}$ |
| RankUp + RDA (Ours) | $\mathbf{5.51}_{\pm0.07}$ | $\mathbf{0.844}_{\pm0.004}$ | $\mathbf{0.890}_{\pm0.003}$ |
| Fully-Supervised | $4.85_{\pm0.01}$ | $0.875_{\pm0.000}$ | $0.910_{\pm0.001}$ |

## A.3 Analysis of Different Semi-Supervised Regression Methods on RankUp

We evaluated the impact of different semi-supervised regression methods on training RankUp's regression output. The ARC was trained using FixMatch, while the regression output was trained with various semi-supervised regression techniques. We compared these results with a supervised setting, where only labeled data was used for training the regression output (denoted as "Supervised"). The results, shown in Table 5, indicate that using different semi-supervised regression methods can further improve performance in MAE, $R^2$, and SRCC metrics compared to the supervised method alone. Demonstrating the effectiveness of leveraging unlabeled data to directly train the regression head.

Among all the semi-supervised regression methods we tested, our proposed RDA achieves the best performance in terms of MAE and $R^2$, with at least a 7.7% improvement in MAE and a 1.7% improvement in $R^2$ compared to other methods. However, it shows a decrease in SRCC compared to MixMatch, with a 1.1% drop. In theory, RDA can also be used alongside the $\Pi$-Model, Mean Teacher, and MixMatch, as it focuses on refining pseudo-labels, which is distinct from the approaches of these methods. However, due to increased computational expense and the goal of maintaining a simpler framework, our proposed RankUp only uses a supervised method or RDA for training the regression output.

Table 5: Comparison of using different semi-supervised regression methods for training RankUp's regression output. Results are evaluated on UTKFace dataset with a setting of 250 labeled samples.

|  | MAE$\downarrow$ | $R^2\uparrow$ | SRCC$\uparrow$ |
|---|---|---|---|
| Supervised | $7.06_{\pm 0.11}$ | $0.751_{\pm 0.011}$ | $0.835_{\pm 0.008}$ |
| $\Pi$-Model | $6.95_{\pm 0.16}$ | $0.758_{\pm 0.010}$ | $0.837_{\pm 0.005}$ |
| Mean Teacher | $7.01_{\pm 0.17}$ | $0.752_{\pm 0.013}$ | $0.831_{\pm 0.004}$ |
| MixMatch | $7.12_{\pm 0.09}$ | $0.769_{\pm 0.006}$ | $\mathbf{0.866}_{\pm 0.002}$ |
| RDA (Ours) | $\mathbf{6.57}_{\pm 0.18}$ | $\mathbf{0.782}_{\pm 0.012}$ | $0.856_{\pm 0.005}$ |

## A.4 Ablation Studies on RankUp Components: ARC and RDA

To further understand the effect of ARC and RDA in RankUp, we conducted ablation studies comparing the performance of a supervised baseline, RDA only, ARC only, and ARC + RDA on the UTKFace dataset, using settings of 50 and 250 labeled samples.

The results, as shown in Table 6, demonstrate that RDA alone does not consistently outperform the supervised baseline, particularly in the 50 labeled samples setting (as reflected in the MAE and $R^2$ values). This may be due to one of the assumptions of RDA not being met, where it requires the pseudo-labels to have reasonably accurate ranking (the supervised baseline with 50 labeled samples only has an SRCC score of 0.371). In contrast, ARC alone ("RankUp" in the table) significantly improves performance over both the supervised baseline and RDA alone. The combination of ARC and RDA yields the best performance, highlighting their synergistic relationship, where it is beneficial to use RDA with ARC.

Table 6: Ablation studies on the ARC and RDA components introduced in RankUp. In this table, "RankUp" refers to the use of the ARC component only, while "RDA" refers to the use of the RDA component only. "RankUp + RDA" refers to the combined use of both ARC and RDA. Results are evaluated on the UTKFace dataset with 50 and 250 labeled samples.

|  | UTKFace (Image Age Estimation) | | | | | |
|---|---|---|---|---|---|---|
|  | Labels = 50 | | | Labels = 250 | | |
|  | MAE $\downarrow$ | R2 $\uparrow$ | SRCC $\uparrow$ | MAE $\downarrow$ | R2 $\uparrow$ | SRCC $\uparrow$ |
| Supervised | $14.13_{\pm 0.56}$ | $0.090_{\pm 0.092}$ | $0.371_{\pm 0.071}$ | $9.42_{\pm 0.16}$ | $0.540_{\pm 0.014}$ | $0.712_{\pm 0.010}$ |
| RDA (Ours) | $14.34_{\pm 1.27}$ | $0.060_{\pm 0.125}$ | $0.442_{\pm 0.104}$ | $8.64_{\pm 0.22}$ | $0.609_{\pm 0.023}$ | $0.772_{\pm 0.012}$ |
| RankUp (Ours) | $9.96_{\pm 0.62}$ | $0.514_{\pm 0.043}$ | $0.703_{\pm 0.019}$ | $7.06_{\pm 0.11}$ | $0.751_{\pm 0.011}$ | $0.835_{\pm 0.008}$ |
| RankUp + RDA (Ours) | $9.33_{\pm 0.54}$ | $0.552_{\pm 0.041}$ | $0.770_{\pm 0.009}$ | $6.57_{\pm 0.18}$ | $0.782_{\pm 0.012}$ | $0.856_{\pm 0.005}$ |

## A.5 Modifications for Adapting USB Codebase for Regression Tasks

The USB [26] codebase is originally designed for semi-supervised classification tasks and includes implementations of various existing semi-supervised classification methods. To adapt it for regression tasks, enabling the implementation of our proposed framework RankUp and other semi-supervised regression methods, we made the following key modifications:

1. Replaced the cross entropy loss function with the MAE loss function.

2. Adjusted the output layer to produce a single continuous output instead of multiple outputs used for multi-class classification.

3. Removed one-hot encoding from the codebase.

4. Normalized the regression labels to the 0-1 range.

5. For methods like Mean Teacher [24] and Π-Model [17], no changes were needed to the core algorithms, as they are inherently applicable to regression tasks.

6. For MixMatch [3], we excluded components specifically designed for classification, such as sharpening and one-hot label encoding, while retaining the input mixing and consistency regularization aspects, which are valuable for regression tasks.

## A.6 Influence of Data Augmentation on Labeled Data

To further analyze the effect of weak augmentation on labeled data, we conducted experiments on the UTKFace dataset with a setting of 250 labeled samples, comparing the results with and without weak augmentation applied to the labeled data.

Table 7 presents a comparison of the results, illustrating the significance of weak augmentation. The performance metrics (MAE, $R^2$, SRCC) show a decline when weak augmentation is not applied (note that weak augmentation was applied to all the baselines in the paper). Despite the observed drop in performance when weak augmentation is not applied, the relative order of effectiveness among the methods remains consistent, regardless of the application of weak augmentation. This consistency further underscores the robustness of our proposed method.

Table 7: Analysis of the effect of weak augmentation on labeled data. Results are evaluated on the UTKFace dataset with a setting of 250 labeled samples. Best results for applying or not applying weak augmentation for each method are highlighted in bold.

| | Weak Augmentation | MAE $\downarrow$ | R2 $\uparrow$ | SRCC $\uparrow$ |
|---|---|---|---|---|
| Supervised | Yes | **9.42**$_{\pm 0.16}$ | **0.540**$_{\pm 0.014}$ | **0.712**$_{\pm 0.010}$ |
| | No | 11.73$_{\pm 0.17}$ | 0.315$_{\pm 0.012}$ | 0.556$_{\pm 0.020}$ |
| Π-Model | Yes | **9.45**$_{\pm 0.30}$ | **0.534**$_{\pm 0.030}$ | **0.706**$_{\pm 0.015}$ |
| | No | 11.97$_{\pm 0.39}$ | 0.319$_{\pm 0.029}$ | 0.567$_{\pm 0.026}$ |
| MixMatch | Yes | **7.95**$_{\pm 0.15}$ | **0.692**$_{\pm 0.013}$ | **0.832**$_{\pm 0.008}$ |
| | No | 10.80$_{\pm 0.09}$ | 0.446$_{\pm 0.021}$ | 0.716$_{\pm 0.003}$ |
| RankUp + RDA (Ours) | Yes | **6.57**$_{\pm 0.18}$ | **0.782**$_{\pm 0.012}$ | **0.856**$_{\pm 0.005}$ |
| | No | 7.16$_{\pm 0.25}$ | 0.742$_{\pm 0.022}$ | 0.843$_{\pm 0.010}$ |
| Fully-Supervised | Yes | **4.85**$_{\pm 0.01}$ | **0.875**$_{\pm 0.000}$ | **0.910**$_{\pm 0.001}$ |
| | No | 5.58$_{\pm 0.02}$ | 0.837$_{\pm 0.002}$ | 0.888$_{\pm 0.000}$ |

## A.7 Data Augmentation Operators

We followed the settings in the USB [26] codebase for augmentation operators, with an adjustment to the strong augmentation for audio data. This adjustment was necessary because our task involves quality assessment, and the original strong augmentation method would have affected the quality of the data. Specifically, we used the following augmentation techniques:

**Image**

- Weak augmentation: Random Crop, Random Horizontal Flip
- Strong augmentation: RandAugment [9]

**Audio**

- Weak augmentation: Random Sub-sample
- Strong augmentation: Random Sub-sample, Random Mask, Random Trim, Random Padding

**Text**

- Weak augmentation: None

- Strong augmentation: Back-Translation [30]

## A.8   Hardware Specifications

All experiments reported in this paper were conducted using an Nvidia Titan XP with 12 GB of VRAM and an Nvidia GeForce RTX 2080 Ti, also equipped with 12 GB of VRAM.

## A.9   More Feature Visualization Results

Additional feature visualization results beyond those presented in this paper are available. This includes t-SNE and UMAP visualizations, in both 2D and 3D, for different semi-supervised regression methods, all accessible at `https://github.com/pm25/semi-supervised-regression`.

## A.10   Dataset Processing

In our experiments, if the dataset provides a pre-defined train-eval-test split, we utilize the training split to train the model and evaluate its performance on the evaluation or test split. If the dataset does not provide such a split, we randomly sample 80% of the data as the training set and the remaining 20% as the test set. We open-source the train-test splits used for conducting the experiments in this paper at `https://github.com/pm25/regression-datasets`.

## A.11   Datasets

Three datasets are utilized in the experiments conducted in this paper: UTKFace [37], BVCC [8], and the Yelp Review [1] dataset. Below, we provide a brief introduction to each dataset.

**UTKFace.** The UTKFace dataset is an image age estimation dataset, where the goal is to predict the age of the person in an image. The labels range from 1 to 116 years old. The dataset consists of 23,705 face images, which we split into 18,964 training samples and 4,741 test samples. The dataset is available in two versions: the original images and an aligned and cropped version. The experiments conducted in this paper use the aligned and cropped version of the UTKFace dataset.

**BVCC.** The VoiceMOS2022 (BVCC) dataset is an audio quality assessment dataset, where the objective is to predict the quality of an audio sample. The labels, ranging from 1 to 5, are obtained by averaging the scores provided by multiple listeners. The dataset is split into training (4,974 samples), evaluation (1,066 samples), and testing (1,066 samples) sets. In the experiments reported in this paper, we utilize only the training and evaluation splits to evaluate performance.

**Yelp Review.** The Yelp Review dataset is a text opinion mining task, where the goal is to predict the rating of customers based on the comments they leave on the Yelp website. There are only five distinct ratings (0 to 4). We use the processed Yelp Review data provided by the USB [26] codebase. The dataset comprises training (250,000 samples), evaluation (25,000 samples), and testing (10,000 samples) sets. We only utilize the training split for model training and the evaluation set for evaluation.

## A.12   Hyperparameter Fine-Tuning

We began by fine-tuning the hyperparameters of the supervised baseline for each dataset, initially setting them based on the configurations provided in the USB [26] codebase. We then fine-tuned the learning rate, weight decay, and layer decay hyperparameters for the supervised baseline model to identify the optimal set of hyperparameters that produced the lowest MAE score. The same hyperparameter settings were subsequently applied to all other methods evaluated in this study. Additionally, each method underwent further fine-tuning by adjusting its own specific additional hyperparameters, distinct from those used in the supervised setting.

### A.13 Hyperparameters

In this section, we list the hyperparameters used in each experimental setting presented in the paper. Table 8 provides the common hyperparameters for the base models: Wide ResNet-28-2 (for image data), Whisper-Base (for audio data), and Bert-Small (for text data). Specific hyperparameter configurations for each semi-supervised regression method are detailed in Table 9. The full code and hyperparameters are open-sourced at `https://github.com/pm25/semi-supervised-regression`.

Table 8: Common hyperparameters for the base models: Wide ResNet-28-2 (for image data), Whisper-Base (for audio data), and Bert-Small (for text data).

|  | Wide ResNet-28-2 | Whisper-Base | Bert-Small |
|---|---|---|---|
| Training Iterations | 262,144 | 102,400 | 102,400 |
| Evaluation Iterations | 1,024 | 1,024 | 1,024 |
| Training Batch Size | 32 | 8 | 8 |
| Optimizer | SGD | AdamW | AdamW |
| Momentum | 0.9 | - | - |
| Criterion | MAE | MAE | MAE |
| Weight Decay | 1e-03 | 2e-05 | 5e-04 |
| Layer Decay | 1.0 | 0.75 | 0.75 |
| Learning Rate | 1e-02 | 2e-06 | 1e-05 |
| EMA Weight | 0.999 | - | - |
| Pretrained | False | True | True |
| Sampler | Random | Random | Random |
| Image Resize | 40x40 | - | - |
| Max Length Seconds | - | 6.0 | - |
| Sample Rate | - | 16,000 | - |
| Max Length | - | - | 512 |

Table 9: Specific hyperparameters for each semi-supervised regression methods.

|  | Π-Model, MeanTeacher | MixMatch | UCVME | CLSS | RankUp | RankUp +RDA |
|---|---|---|---|---|---|---|
| Unlabeled Batch Ratio | 1.0 | 1.0 | 1.0 | 0.25 | 7.0 | 7.0 |
| Regression Unlabeled Loss Ratio | 0.1 | 0.1 | 0.05 | - | - | 1.0 |
| Regression Unlabeled Loss Warmup | 0.4 | 0.4 | - | - | - | 0.4 |
| Mixup Alpha | - | 0.5 | - | - | - | - |
| Dropout Rate | - | - | 0.05 | - | - | - |
| Ensemble Number | - | - | 5 | - | - | - |
| CLSS Lambda | - | - | - | 2.0 | - | - |
| Labeled Contrastive Loss | - | - | - | 1.0 | - | - |
| Unlabeled Contrastive Loss | - | - | - | 0.05 | - | - |
| Unlabeled Rank Loss Ratio | - | - | - | 0.01 | - | - |
| ARC Unlabeled Loss Ratio | - | - | - | - | 1.0 | 1.0 |
| ARC Loss Ratio | - | - | - | - | 0.2 | 0.2 |
| Confidence Threshold | - | - | - | - | 0.95 | 0.95 |
| Temperature | - | - | - | - | 0.5 | 0.5 |
| RDA Refinement Steps | - | - | - | - | - | 1,024 |

