# OpenReview forum: "RankUp: Boosting Semi-Supervised Regression with an Auxiliary Ranking Classifier"
_NeurIPS.cc/2024/Conference — NeurIPS 2024 poster_

### Official Review · Reviewer_Ngei · 2024-07-11

**Soundness:** 3
**Presentation:** 3
**Contribution:** 3
**Rating:** 6
**Confidence:** 3

**Summary:**

The submission presents a new semi-supervised learning algorithm for deep regression models: in addition to the standard supervised regression head, an auxiliary classification head is trained using a semi-supervised learning algorithm for classification that learns to classify pairs of examples. A pair is classified correctly if the example with the larger target value is ranked above the example with the smaller target value. The loss for supervised regression is used in a weighted sum with the loss for semi-supervised classification. A further improvement to the algorithm is obtained by introducing an additional component in the loss that encourages the distribution of predicted target values to be aligned with the distribution of ground-truth target values. Experiments on three regression problems, one involving images, one involving audio, and one involving text, indicate that the proposed method outperforms semi-supervised regression algorithms from the literature.

**Strengths:**

The submission presents a nifty idea for semi-supervised regression: formulate a label ranking problem using the regression task at hand to be able to apply a semi-supervised learning algorithm for classification. It additionally introduces a distribution alignment loss for regression that yields further improvements in performance.

**Weaknesses:**

The only potential problem I see is the influence of augmentation on the results. The FixMatch method used to train the auxiliary classifier, shown in Algorithm 1, applies weak augmentation for the supervised component of the loss. For a fair comparison, it seems crucial to apply this weak augmentation also on the labeled data for all the other semi-supervised and purely supervised configurations evaluated in the empirical comparison. The submission does not state whether this was done.

Another, smaller issue is that the submission does not state exactly which operators were used for augmentation in the three different domains.

My negative ratings for soundness, contribution, and the overall quality are based purely on this concern regarding the use of augmentation. *** I HAVE INCREASED BY RATINGS AFTER THE REBUTTAL ***

**Questions:**

Was weak augmentation on the labeled data applied consistently across all the semi-supervised and supervised learning algorithm configurations evaluated in the comparison?

Which forms of augmentation were applied in the three different domains?

**Limitations:**

The submission discusses limitations of the proposed approach.

---

> ### Author Rebuttal · Authors · 2024-08-06
>
> Thank you for your thoughtful review and highlighting the strengths of our approach. We appreciate your careful consideration, particularly regarding augmentation. We'd like to address your concerns and questions:
>
> ### **W1) Influence of augmentation on results**
> We confirm that **weak augmentation was applied** consistently to the labeled data across **all** semi-supervised and supervised learning algorithm configurations in our evaluation. This ensures a fair comparison between our proposed method and the baselines, ensuring that performance improvements are not due to differences in augmentation strategies. We will include these details in the revised paper for better clarity. Our codes along with implementation details will be published if the paper is accepted.
>
> To further analyze the effect of weak augmentation, we conducted a **new ablation study** on the UTKFace dataset with 250 labeled samples, **without** weak augmentation applied to the labeled data. Below is a comparison of the results:
>
> |  |  |  |  |  |
> |:----------------------|:------------------:|:--------------:|:-----------:|:----------:|
> |                 | Weak Augmentation |     MAE ↓      |   $R^2$ ↑   |   SRCC ↑   |
> | Supervised            | Yes                | 9.42±0.16      | 0.540±0.014 | 0.712±0.010|
> | **Supervised**        | **No**             | **11.73±0.17** | **0.315±0.012** | **0.556±0.020** |
> | $\pi$ Model           | Yes                | 9.45±0.30      | 0.534±0.030 | 0.706±0.015|
> | **$\pi$ Model**       | **No**             | **11.97±0.39**      | **0.319±0.029**        | **0.567±0.026**       |
> | MixMatch              | Yes                | 7.95±0.15      | 0.692±0.013 | 0.832±0.008|
> | **MixMatch**          | **No**             | **10.80±0.09**      | **0.446±0.021**        | **0.716±0.003**       |
> | RankUp + RDA (Ours)          | Yes                | 6.57±0.18      | 0.782±0.012 | 0.856±0.005|
> | **RankUp + RDA (Ours)**      | **No**             | **7.16±0.25**      | **0.742±0.022**        | **0.843±0.010**       |
> | Fully-Supervised      | Yes                | 4.85±0.01      | 0.875±0.000 | 0.910±0.001|
> | **Fully-Supervised**  | **No**             | **5.58±0.02**  | **0.837±0.002** | **0.888±0.000** |
> |  |  |  |  |  |
>
> This table illustrates the importance of weak augmentation. The performance metrics (MAE, $R^2$, SRCC) decrease when weak augmentation is not applied **(please note that weak augmentation was applied on all the baselines in the paper)**. Despite this drop, the order of effectiveness among the methods remains consistent, whether weak augmentation is used or not. This consistency in ranking highlights the robustness of our proposed method.
>
> ### **W2) Lack of specificity regarding augmentation operators**
> We followed the settings in the USB \[1\] codebase for augmentation operators, with an adjustment to the strong augmentation for audio data. This adjustment was necessary because our task involves quality assessment, and the original strong augmentation method would have affected the quality of the data. Specifically, we used the following augmentation techniques:
>
> 1. Image:
>    * Weak augmentation:  Random Crop, Random Horizontal Flip
>    * Strong augmentation: RandAugment \[2\]
> 2. Audio:
>    * Weak augmentation: Random Sub-sample
>    * Strong augmentation: Random Sub-sample, Random Mask, Random Trim, Random Padding
> 3. Text:
>    * Weak augmentation: None
>    * Strong augmentation: Back-Translation \[3\]
>
> ### **Q1) Consistent application of weak augmentation**
> Yes, weak augmentation on the labeled data **was applied consistently** across all semi-supervised and supervised learning algorithm configurations in our evaluation. More information can be found in the response to W1.
>
> ### **Q2) Forms of augmentation in different domains**
> As detailed above (W2), we used domain-specific augmentation techniques based on the USB \[1\] codebase, with an adjustment for audio data.
>
> We appreciate you bringing these important points to our attention. Addressing these issues will improve the clarity and reproducibility of our work. We will revise our manuscript to incorporate these details.
>
> ---
>
> \[1\] Wang, Yidong, et al. "Usb: A unified semi-supervised learning benchmark for classification." *Advances in Neural Information Processing Systems* 35 (2022): 3938-3961.
>
> \[2\] Cubuk, Ekin D., et al. "Randaugment: Practical automated data augmentation with a reduced search space." *Proceedings of the IEEE/CVF conference on computer vision and pattern recognition workshops*. 2020.
>
> \[3\] Xie, Qizhe, et al. "Unsupervised data augmentation for consistency training." *Advances in neural information processing systems* 33 (2020): 6256-6268.

---

> > ### Comment · Reviewer_Ngei · 2024-08-08
> > **Response to rebuttal**
> >
> > Thank you for your rebuttal. My concerns have been addressed, and I will raise my score accordingly.

---

> > > ### Author Response · Authors · 2024-08-09
> > >
> > > Thank you for your thoughtful review and positive consideration!

---

### Official Review · Reviewer_tpBB · 2024-07-12

**Soundness:** 3
**Presentation:** 2
**Contribution:** 4
**Rating:** 6
**Confidence:** 4

**Summary:**

This paper presents a simple yet effective approach that adapts existing semi-supervised classification techniques to enhance the performance of regression tasks. The perspective is novel and can effectively use existing technologies to solve regression problems.

**Strengths:**

The article has novel ideas and solid experimental results.

**Weaknesses:**

The method description needs to be improved.

**Questions:**

1. The quality of the figures in the article needs to be improved.
2. In the method, several variables are unclear. For example, ℓarc, ℓregression, etc.
3. The description of some processes in Section 3.3 is not very clear and needs to be improved.

**Limitations:**

as above

---

> ### Author Rebuttal · Authors · 2024-08-06
>
> Thank you for your positive comments on the novelty of our ideas and the strength of our experimental results. We appreciate your constructive feedback and will address your concerns as follows:
>
> ### **W1) The method description needs to be improved**
> We will thoroughly revise Section 3 to enhance clarity by:
>
> 1. A more detailed step-by-step explanation of our approach
> 2. Clearer definitions of all variables and components
> 3. Illustrative examples to aid understanding
> 4. Revise the unclear sentences.
>
> ### **Q1) The quality of the figures in the article needs to be improved.**
> We have thoroughly improved the quality of figures by:
>
> 1. Increasing the resolution of all figures
> 2. Enlarging the font size in all figures
> 3. Changing the font color to black for better readability
> 4. Adding more informative captions and labels
> 5. Unifying the size of subfigures in Figure 3
>
> The revised figures are included in the uploaded PDF in the global response section for your review.
>
> ### **Q2) Several variables are unclear. For example, ℓarc, ℓregression, etc.**
> We will provide clear definitions for all variables in the revised manuscript. To address the specific examples you mentioned:
>
> 1. ℓarc represents the loss of the auxiliary ranking classifier (ARC).
> 2. ℓregression denotes the regression loss.
>
> ### **Q3) The description of some processes in Section 3.3 is not very clear and needs to be improved**
>
> We will thoroughly revise Section 3.3 to improve clarity by:
>
> 1. Providing a more detailed explanation of each process
> 2. Using consistent terminology throughout the section
> 3. Adding a step-by-step algorithm or flowchart to illustrate the processes
> 4. Including examples to demonstrate how each process works in practice
>
> Thank you again for your constructive feedback. We believe these revisions will significantly improve the clarity and readability of our paper, making our method and contributions more accessible to readers.

---

> > ### Comment · Reviewer_tpBB · 2024-08-09
> >
> > Thank you for your rebuttal. I have no quetions.

---

> > > ### Author Response · Authors · 2024-08-12
> > >
> > > Thank you for your thoughtful review and positive feedback!

---

### Official Review · Reviewer_41bS · 2024-07-13

**Soundness:** 3
**Presentation:** 3
**Contribution:** 2
**Rating:** 4
**Confidence:** 4

**Summary:**

The authors present two components to improve the problem of semi-supervised regression - 1) RankUp which considers the regression problem as a ranking problem and then adapts existing semi-supervised classification methods, and 2) Regression distribution alignment (RDA) which is to refine pseudo-labels. The experiments demonstrate the effectiveness on the problem.

**Strengths:**

- The manuscript is clear and easy to follow;
- The method is technically sound;
- The experimental demonstration shows effectiveness over other baselines.

**Weaknesses:**

I'm concerned about the technical novelty.

  The proposed RankUp and RDA are combination from existing methods [1][2][3], and the core idea of adding ranking auxiliary objective to the regression object is not novel [4][5]. The author should include more discussion on the related work of using ranking auxiliary objective to improve regression problems.

  The two proposed components seem to be orthogonal, especially for RankUp is directly designed to improving semi-supervised problems, but for more general regression tasks.

[1] Burges et al., Learning to rank using gradient descent. ICML 2005

[2] Sohn et al., Fixmatch: Simplifying semi-supervised learning with consistency and confidence, NeurIPS 2020

[3] Kim et al., Distribution aligning refinery of pseudo-label for imbalanced semi-supervised learning, NeurIPS 2020

[4] Rank-N-Contrast: Learning Continuous Representations for Regression, NeurIPS 2023

[5] Gong et al., RankSim: Ranking Similarity Regularization for Deep Imbalanced Regression, ICML 2022

**Questions:**

To better understand the effectiveness of each component on the semi-supervised regression, it can be better if the authors add more discussion and ablation studies on RankUp and RDA individually - e.g. only with RDA.

---

> ### Author Rebuttal · Authors · 2024-08-06
>
> Thank you for your thoughtful review and for bringing these important points to our attention. We appreciate your feedback on the clarity and technical soundness of our manuscript. Here are our responses to your concerns and questions:
>
> ### **W1)  Regarding concerns about technical novelty**
> We acknowledge that RankUp and RDA are developed based on existing methods. However, our work uniquely integrates and adapts these methods for **semi-supervised regression tasks.**  Below, we highlight the key differences between our work and the relevant existing methods:
>
> **\[1\] Burges et al., Learning to rank using gradient descent. ICML 2005:** While RankNet uses pairwise ranking loss for ranking the data, the original study focuses on information retrieval tasks. We adapt the concept of RankNet for semi-supervised regression tasks, transforming the regression problem into a classification problem. This allows us to apply existing **semi-supervised** classification models to **regression problems**, bridging a gap in the field.
>
> **\[2\] Sohn et al., FixMatch: Simplifying semi-supervised learning with consistency and confidence, NeurIPS 2020:** FixMatch is a popular method for semi-supervised classification tasks. Our work first verified that FixMatch can also be used for semi-supervised **regression tasks.** Additionally, we would like to point out that we adopted FixMatch as a representative method to train RankUp's auxiliary ranking classifier. Our framework is flexible and can incorporate various semi-supervised classification methods, not just FixMatch.
>
> **\[3\] Kim et al., Distribution aligning refinery of pseudo-label for imbalanced semi-supervised learning, NeurIPS 2020:**
>    1. Similarity: Align the pseudo-labels distribution with the labeled data distribution.
>    2. Difference: DARP can’t be applied to regression tasks, and thus we designed a fundamentally different approach called RDA that can be used for **regression tasks**. Additionally, we introduced techniques to **overcome computational bottlenecks** for RDA, as detailed in Section 3.3 of our paper.
>
> **\[4\] Rank-N-Contrast: Learning Continuous Representations for Regression, NeurIPS 2023:**
>    1. Similarity: Uses ranking loss in regression tasks.
>    2. Difference: Rank-N-Contrast is designed for supervised settings and can’t directly apply to semi-supervised settings. The focus of our study is **semi-supervised regression tasks**. Rank-N-Contrast requires the true labels of the data, whereas in a semi-supervised setting, the majority of the data is unlabeled. Moreover, one of the motivations of our proposed method is to leverage existing semi-supervised classification techniques for regression tasks, which Rank-N-Contrast can't achieve since it's a contrastive learning approach, not a classification approach.
>
> **\[5\] Gong et al., RankSim: Ranking Similarity Regularization for Deep Imbalanced Regression, ICML 2022:**
>    1. Similarity: Uses ranking loss in regression tasks.
>    2. Difference: RankSim is similar to Rank-N-Contrast, which can't directly apply to semi-supervised settings. It requires the true labels of the data, whereas in a semi-supervised setting, the majority of the data is unlabeled. More detail can reference the above response to Rank-N-Contrast.
>
> We will expand our related work section to provide a more comprehensive discussion of existing ranking-based regression methods.
>
> ### **W2) Interconnection between RankUp and RDA**
> Although RankUp and RDA may appear unrelated at first glance, they are suitably interconnected in our semi-supervised regression framework:
>
> 1. One of the key assumptions of RDA is reliant on the ranking of the pseudo-labels. The better the quality of the ranking of pseudo-labels, the better RDA will perform.
> 2. RankUp enhances the ranking of pseudo-labels by incorporating ranking information.
> 3. The enhanced pseudo-label quality from RankUp directly benefits RDA's effectiveness, creating a synergistic relationship.
>
> ### **Q1) Add more discussion and ablation studies on RankUp and RDA**
>
> We appreciate your suggestion for additional ablation studies. Here are the results comparing supervised learning, RDA only, RankUp only, and RankUp \+ RDA on the UTKFace dataset:
>
> |  |  | Labels \= 50 |  |  |  | Labels \= 250 |  |
> | :---: | :---: | :---: | :---: | :---: | :---: | :---: | :---: |
> |  | MAE &#8595; | $R^2$ &#8593; | SRCC &#8593; | | MAE &#8595; | $R^2$ &#8593; | SRCC &#8593; |
> | Supervised | 14.13±0.56 | 0.090±0.092 | 0.371±0.071 | | 9.42±0.16 | 0.540±0.014 | 0.712±0.010 |
> | **RDA (Ours)** | **14.34±1.27** | **0.060±0.125** | **0.442±0.104** | | **8.64±0.22** | **0.609±0.023** | **0.772±0.012** |
> | RankUp (Ours) | 9.96±0.62 | 0.514±0.043 | 0.703±0.019 | | 7.06±0.11 | 0.751±0.011 | 0.835±0.008 |
> | RankUp \+ RDA (Ours) | 9.33±0.54 | 0.552±0.041 | 0.770±0.009 | | 6.57±0.18 | 0.782±0.012 | 0.856±0.005 |
>
> These results demonstrate that:
>
> 1. RDA alone may not improve upon the supervised baseline (MAE and $R^2$ of labeled = 50 setting).
> 2. RankUp significantly outperforms the baseline and RDA alone.
> 3. The combination of RankUp and RDA yields the best performance, showcasing their synergistic relationship.
>
> We will include this ablation study and a detailed discussion of the results in our revised manuscript. This will provide a clearer understanding of each component's contribution and their combined effect on semi-supervised regression performance.
>
> Thank you again for your insightful comments. We believe addressing these points will significantly strengthen our paper and provide a more comprehensive understanding of our method's contributions to semi-supervised regression.

---

> > ### Author Response · Authors · 2024-08-12
> >
> > Thank you very much for your time and effort in reviewing our work. We have addressed your comments and welcome any further feedback or questions you may have. We would be happy to discuss them with you.

---

### Official Review · Reviewer_PMad · 2024-07-15

**Soundness:** 3
**Presentation:** 3
**Contribution:** 3
**Rating:** 6
**Confidence:** 3

**Summary:**

The work introduces a novel SSL-regression method called RankUp. A a pairwise ranking loss enables the SSL-method FixMatch to utilize also the unlabled split of the data to learn a regression task. The addition of the Regression Distribution Alignment (RDA) loss enables the method to also take the overall distribution of the labeled samples into account. RankUp achieves SOTA on multiple SSL regression benchmarks.

**Strengths:**

The paper is well written and easy to follow. Although simple, the proposed  combination of pseudo-label based SSL approach FixMatch with a pairwise ranking loss is novel. The effect of RDA is plausible and its limitations are discussed. Both variants report promising performance gains in small scale regression benchmarks, especially when the number of labeled samples is small. The qualitative analysis via t-SNE visualization underlines the effect of RankUp on the learned  representation.

**Weaknesses:**

The scale of the experiments. Although this might be an issue of the SSL-regression domain in general.

**Questions:**

Regarding the baseline methods: "Specifically, we adapt the popular semi-supervised learning codebase USB [20], modifying it for regression tasks." What modifications are necessary to turn e.g. the Mean Teacher approach into a regression method?

**Limitations:**

The authors have adequately addressed the limitations.

---

> ### Author Rebuttal · Authors · 2024-08-05
>
> Thank you for your positive comments and insightful questions. We're pleased you found our approach novel and well-supported. Regarding the weakness and questions:
>
> ### **W1) Scale of the Experiments**
>
> We acknowledge that the scale of our experiments is limited, which is indeed a common challenge in semi-supervised learning for regression tasks. Traditionally, research in this area has focused on small-scale image datasets \[1\]\[2\]. To address this limitation and enhance the robustness of our findings, we have evaluated our methods not only on the image datasets but also across diverse datasets including image, audio, and text modalities. These varied experiments consistently demonstrated promising results, reinforcing the efficacy of our approach.
>
> ### **Q1) Modifications for Adapting USB Codebase for Regression Tasks**
>
> The USB \[3\] codebase is designed for semi-supervised classification tasks. To adapt it for regression tasks, we made the following modifications:
>
> 1. We replaced the softmax loss function with the mean absolute error (MAE) loss function.
> 2. We adjusted the output layer to produce a single continuous output instead of multiple outputs used for multi-class classification.
> 3. For methods like Mean Teacher and $\pi$ Model, we didn't need to change the core algorithms as they are inherently applicable to regression tasks.
> 4. For MixMatch, we excluded components specifically designed for classification, such as sharpening and one-hot label encoding. We retained the input mixing and consistency regularization aspects, as these are valuable for regression tasks as well.
>
> We appreciate you highlighting the need for clarity on these modifications. We will include these details in the final paper to provide a clearer understanding of our experiments.
>
> ---
>
> \[1\] Dai, Weihang, Xiaomeng Li, and Kwang-Ting Cheng. "Semi-supervised deep regression with uncertainty consistency and variational model ensembling via bayesian neural networks." *Proceedings of the AAAI Conference on Artificial Intelligence*. Vol. 37\. No. 6\. 2023\.
>
> \[2\] Dai, Weihang, et al. "Semi-supervised contrastive learning for deep regression with ordinal rankings from spectral seriation." *Advances in Neural Information Processing Systems* 36 (2023): 57087-57098.
>
> \[3\] Wang, Yidong, et al. "Usb: A unified semi-supervised learning benchmark for classification." *Advances in Neural Information Processing Systems* 35 (2022): 3938-3961.

---

> > ### Comment · Reviewer_PMad · 2024-08-12
> >
> > I thank the authors for their rebuttal. I keep my positive score.

---

> > > ### Author Response · Authors · 2024-08-12
> > >
> > > Thank you for your thoughtful review and positive feedback!

---

### Author Rebuttal · Authors · 2024-08-06

We would like to thank all the reviewers for taking their time to review our work. We greatly appreciate the thoughtful feedback and insightful comments, which have significantly contributed to improving the quality of our paper. We have attached the revised figures in the PDF for the reference.

---

### Author Response · Authors · 2024-08-14

Dear Area Chair,

We would like to provide a summary following the authors' rebuttal. We sincerely thank all the reviewers for their thorough evaluations and valuable feedback. We are especially grateful to Reviewer Ngei for the opportunity to clarify potential issues and for being willing to update the rating **from 3 to 6**. We also appreciate Reviewers PMad and tpBB for recognizing our paper's technical soundness and providing **positive ratings (6)** prior to the rebuttal.

While we thank Reviewer 41bS for the thoughtful review, we note that they provided lower scores and raised several questions. We have carefully addressed the reviewer's comments in our rebuttal but have not received any feedback in response. As a result, we haven't had the opportunity to engage in further discussion with the reviewer. Nevertheless, we are confident that we have **thoroughly addressed their concerns**.

Regarding the technical novelty concern raised by Reviewer 41bS, which mentions that some existing works have incorporated ranking loss in regression tasks, we would like to clarify the following:

* These methods are designed for **supervised** settings and cannot be applied to **semi-supervised** settings.
* Additionally, these methods are based on contrastive learning approaches, which **do not align** with the objective of our paper. Our work aims to leverage existing semi-supervised classification methods for regression tasks.
* Ranking loss is only one component of our proposed method; we have also introduced **RDA**, which is **highly beneficial** to semi-supervised regression when combined with our proposed RankUp.

More details are available in our rebuttal response to Reviewer 41bS.

According to the input we provided, most reviewers indicated that their concerns were addressed, leading to positive ratings. We sincerely appreciate the reviewers for recognizing the innovation and potential impact of our work.

Thank you\!

Sincerely,
The Authors

---

### Decision · Program_Chairs · 2024-09-25

**Decision:**

Accept (poster)

**Comment:**

This paper modifies existing semi-supervised classifications setups for regression. The reviews raised several issues, most significant among them was novelty. During rebuttal and discussions the authors satisfactorily addressed most of the concerns. This paper improves the performance of regression tasks by converting them into ranking problem.